# SOFA score performs worse than age for predicting mortality in patients with COVID-19

**Raphael A. G. Sherak** [1]*, **Hoomaan Sajjadi** [2], **Naveed Khimani** [2], **Benjamin Tolchin** [3,4], **Karen Jubanyik** [1], **R. Andrew Taylor** [1], **Wade Schulz** [5,6], **Bobak J. Mortazavi** [2,7], **Adrian D. Haimovich** [1,8]

**1** Yale Department of Emergency Medicine, Yale School of Medicine, New Haven, CT, United States of America, **2** Department of Computer Science and Engineering, Center for Remote Health Technologies and Systems, Texas A&M Univ, College Station, TX, United States of America, **3** Department of Neurology, Yale School of Medicine, New Haven, CT, United States of America, **4** Yale New Haven Health Center for Clinical Ethics, New Haven, CT, United States of America, **5** Department of Laboratory Medicine, Yale School of Medicine, New Haven, CT, United States of America, **6** Biomedical Informatics and Data Science, Yale School of Medicine, New Haven, CT, United States of America, **7** Center for Outcomes Research and Evaluation, Yale University, New Haven, CT, United States of America, **8** Department of Emergency Medicine, Beth Israel Deaconess Medical Center, Boston, MA, United States of America

* Raphael.Sherak@Yale.edu

**Data Availability Statement:** Data cannot be shared publicly because of legal and ethical restrictions regarding patient privacy. Data are available from Yale's chief privacy officer, Susan

## Abstract

The use of the Sequential Organ Failure Assessment (SOFA) score, originally developed to describe disease morbidity, is commonly used to predict in-hospital mortality. During the COVID-19 pandemic, many protocols for crisis standards of care used the SOFA score to select patients to be deprioritized due to a low likelihood of survival. A prior study found that age outperformed the SOFA score for mortality prediction in patients with COVID-19, but was limited to a small cohort of intensive care unit (ICU) patients and did not address whether their findings were unique to patients with COVID-19. Moreover, it is not known how well these measures perform across races. In this retrospective study, we compare the performance of age and SOFA score in predicting in-hospital mortality across two cohorts: a cohort of 2,648 consecutive adult patients diagnosed with COVID-19 who were admitted to a large academic health system in the northeastern United States over a 4-month period in 2020 and a cohort of 75,601 patients admitted to one of 335 ICUs in the eICU database between 2014 and 2015. We used age and the maximum SOFA score as predictor variables in separate univariate logistic regression models for in-hospital mortality and calculated area under the receiver operator characteristic curves (AU-ROCs) and area under precision-recall curves (AU-PRCs) for each predictor in both cohorts. Among the COVID-19 cohort, age (AU-ROC 0.795, 95% CI 0.762, 0.828) had a significantly better discrimination than SOFA score (AU-ROC 0.679, 95% CI 0.638, 0.721) for mortality prediction. Conversely, age (AU-ROC 0.628 95% CI 0.608, 0.628) underperformed compared to SOFA score (AU-ROC 0.735, 95% CI 0.726, 0.745) in non-COVID-19 ICU patients in the eICU database. There was no difference between Black and White COVID-19 patients in performance of either age or SOFA Score. Our findings bring into question the utility of SOFA score-based resource allocation in COVID-19 crisis standards of care.

Bouregy, PhD (susan.buregy@yale.edu), for researchers who meet the criteria for access to confidential data. The eICU database is publicly accessible. Pollard, T., Johnson, A., Raffa, J., Celi, L. A., Badawi, O., & Mark, R. (2019). eICU Collaborative Research Database (version 2.0). PhysioNet. https://doi.org/10.13026/C2WM1R.

**Funding:** The authors did not receive any specific funding (financial, material or otherwise) for this study.

**Competing interests:** The authors have declared that no competing interests exist.

## Introduction

The Sequential Organ Failure Assessment (SOFA) score was developed in 1994 by a European Society of Intensive Care Medicine working group to objectively quantify the degree of organ dysfunction and failure in intensive care unit (ICU) patients with sepsis [1]. The SOFA score assigns a value of 0 to 4 to the dysfunction of 6 organ systems (respiratory, coagulation, hepatic, cardiovascular, neurologic, and renal), with higher numbers indicating more dysfunction. The score is calculated using the worst clinical values observed in the previous 24 hours. It may be used to describe each organ system or as a summative measure.

Although the initial intention of the SOFA score was to describe morbidity in ICU patients with sepsis, subsequent studies have used the SOFA score or SOFA score-based models to predict mortality [2–4]. While the SOFA score on admission performed well for mortality prediction in sepsis in some studies [5,6], other studies have suggested only intermediate discriminatory accuracy [7,8]. The use of SOFA score has also expanded beyond patients with sepsis [9,10] and is used outside of the ICU [6,11].

With COVID-19 positive patients occupying up to 90% of ICU beds during the global SARS-CoV-2 pandemic [12], the SOFA score gained new applications. One study reported that 20 of the 26 COVID-19 ventilator triage policies surveyed used the SOFA score [13]. Many of these guidelines for crisis standards of care involve withholding resources from patients with an expected low likelihood of survival, with the rationale of saving the most lives possible with the limited resources available. One way this is done is by assuming patients with a SOFA score at or above a predetermined threshold are unlikely to survive the hospitalization. However, there is a dearth of data about the use of the SOFA score to predict mortality in COVID-19, potentially leading to improper allocation of resources [14,15].

One prominent study of 675 COVID-19 positive patients from a single healthcare system found that SOFA score had inferior discriminant prognostic accuracy for in-hospital mortality compared to patient age [16]. However, the study was limited by its small sample size with missing data for approximately 25% of its cohort and restriction to patients requiring mechanical ventilation. Moreover, the study population had higher and less variable SOFA scores than the average ICU population [6,7].

Here, we expand on prior work of SOFA score mortality prediction in COVID-19 by assessing SOFA score performance for an undifferentiated population of admitted patients. The primary aim of this retrospective study was to assess how well SOFA score predicts in-hospital mortality in a consecutive cohort of patients with COVID-19 admitted to a quaternary medical center in the Northeast United States (U.S.). We compare the discriminative performance of SOFA to age alone. To examine whether any findings are specific to patients with COVID-19, we contrast these findings to a large cohort of general ICU patients from the publicly accessible eICU database. As a secondary objective, given concerns that prediction models can perpetuate systemic inequities [17–22], we compared the prognostic value of SOFA across race in both cohorts.

## Methods

This was a retrospective study comprising two separate patient cohorts.

### Data acquisition and preprocessing

**COVID-19 cohort.** Patients in the COVID-19 cohort consisted of all patients with an age ≥ 18 admitted to any of the 5 hospitals in the Yale-New Haven Health System (YNHH) from March 29th, 2020 to August 1, 2020 with a diagnosis of COVID-19, defined as either a positive PCR test for COVID-19 or designated as a COVID-19 patient by an attending

physician. Data was obtained retrospectively from the YNHH electronic medical record (EMR, Epic Systems Corporation, Verona, WI). SOFA score was automatically calculated for all admitted patients and recorded every 4 hours by the YNHH EMR. Patients were excluded if they did not have a SOFA score recorded in the first 24 hours of admission or if they had a prior admission with a COVID-19 diagnosis. All records were de-identified prior to analysis. The patient cohort is described extensively in a prior, publicly available manuscript [23].

**eICU cohort.** The eICU Collaborative Research Database is a publicly available database containing data from over 139,000 patients hospitalized between 2014 and 2015 at one of the 335 American ICUs participating Philips Healthcare eICU program [24].

Patients with an ICU stay of at least 24 hours were included. Also, for the patients having more than one hospitalization, only the most recent hospitalization was used. The SOFA scores on eICU were calculated and extracted using Python's Pandas library, based on its standard definition on the values of creatinine, bilirubin, platelets, fraction of inspired oxygen, partial pressure of inspired oxygen, Glasgow Coma Scale, mean arterial pressure, and mechanical ventilation status. As part of the eICU patient de-identification process, all patients with age >89 are grouped together and were assigned an age of 90 for the analysis. Then, the corresponding tables and columns were pre-processed to calculate the final scores. The overall steps taken in this procedure are described in the supplement.

In both cohorts, gender was dichotomized and race was classified as a single category based on predefined fields in the EMR. Depending on a patient's clinical status, gender and race were either self-selected by patients or assigned by hospital registration staff.

**Ethics statement.** This study was reviewed and approved by the Yale University Institutional Review Board (Study Number 2000027747). The study was deemed exempt from the requirement for consent because all data were analyzed anonymously.

## Logistic regression model development

Age in years at time of admission and the maximum SOFA score recorded in the first 24 hours after admission were each used as predictor variables in separate univariate logistic regression models for the binary outcome of in-hospital mortality. Each model was fit to a sample of 60% of the respective cohorts.

## Model assessment

The remaining 40% of each cohort were used as a validation cohort to calculate area under the receiver operator characteristic curves (AU-ROCs). AU-ROC intervals were estimated using DeLong's method [25]. This method was then repeated to calculate area under precision-recall curves (AU-PRCs). Calibration curves were produced using R's predtools package [26] (S1 Fig). In addition, we conducted a decision curve analysis [27] using the dcurves R package [28]. All analyses were performed in Python (Version 3.7.7) and R (Version 1.4.1717).

We performed several secondary analyses. We stratified each cohort by race including only patients identifying as Black/African American or White/Caucasian as we had insufficient sample sizes for further groups. Because the eICU cohort is restricted to patients admitted to the ICU, we performed an additional analysis of the COVID-19 cohort restricted to ICU patients. We also conducted an exploratory analysis of the COVID-19 cohort of survival rates at and above a given SOFA score. We then stratified by race to explore whether there was a difference in survival rates at various SOFA score thresholds. The binomial exact test was used to calculate confidence intervals.

## Results

### Patient characteristics

A total of 2,648 patients were included in the COVID-19 cohort with a median age of 65 (IQR 50–79), mean SOFA score of 2.1 (SD 2.7), median SOFA score of 1 (IQR 0–3) and an in-hospital mortality rate of 17.1%. The eICU cohort consisted of 75,601 patients with a median age of 65 (IQR 52–77), mean SOFA score of 7.7 (SD 2.7), median SOFA score of 8 (IQR 6–9) and an in-hospital mortality rate of 10.8%. The COVID-19 and eICU cohorts respectively were 52.4% and 46.1% female and 25.4% and 10.7% Black/African American (Fig 1 and Table 1).

### Primary outcome

Among the COVID-19 cohort, age (AU-ROC 0.795, 95% CI 0.762, 0.828) had a significantly better discrimination than SOFA score (AU-ROC 0.679, 95% CI 0.638, 0.721) for mortality prediction (Fig 2). When restricted to patients admitted to the ICU, age (AU-ROC 0.698, 95% CI 0.631, 0.765) still generally performed better than SOFA score (AU-ROC 0.562, 95% CI 0.486, 0.639), for mortality prediction in COVID-19 patients (S2 Fig). Conversely, age (AU-ROC 0.628 95% CI 0.608, 0.628) significantly underperformed compared to SOFA score (AU-ROC 0.735, 95% CI 0.726, 0.745) in non-COVID-19 eICU patients (Fig 2 and Table 2). These findings were consistent with the decision curve analysis where the age-based model had a higher net benefit across most threshold probabilities in the COVID-19 cohort, while the SOFA score-based model had a higher net benefit than the age-based model across all threshold probabilities greater than 5% in the eICU cohort (Fig 3).

### Secondary analyses

When stratified by race, age performed similarly between Black and White patients in both cohorts (Fig 4 and Table 3). There also was no significant difference in the performance of the

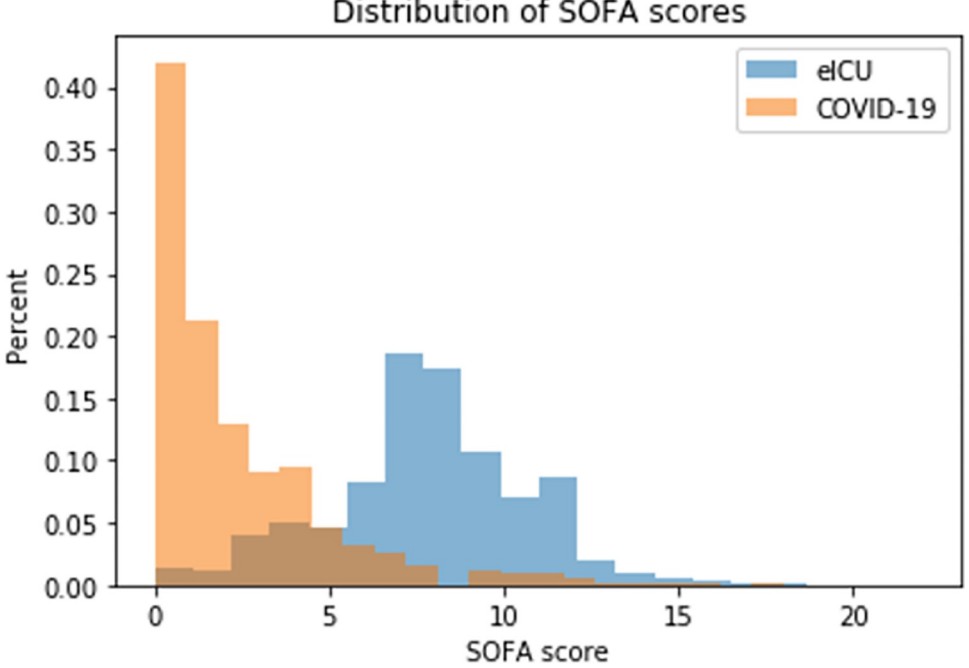

**Fig 1. Distribution of SOFA scores.** Histogram of SOFA scores in the COVID-19 (orange) and eICU (blue) cohorts.

**Table 1. Demographics for COVID-19 and eICU cohorts.**

| COVID-19 Cohort | | Missing | Total | eICU Cohort | | Missing | Total |
|---|---|---|---|---|---|---|---|
| n | | | 2648 | n | | | 75601 |
| Survival to Hospital Discharge, n (%) | | 0 | 2196 (82.9) | Survival to Hospital Discharge, n (%) | | 0 | 67463 (89.2) |
| SOFA Score, median (IQR) | | 0 | 1 (0–3) | SOFA Score, median (IQR) | | 0 | 8 (6–9) |
| Gender, n (%) | Female | 0 | 1387 (52.4) | Gender, n (%) | Female | 12 | 34849 (46.1) |
| | Male | | 1261 (47.6) | | Male | | 40740 (53.9) |
| Age, median (IQR) | | 0 | 65 (50–79) | Age, median (IQR) | | | 65 (52–77) |
| ICU, n (%) | | 0 | 642 (24.2) | ICU, n (%) | | 0 | 75601 (100) |
| Race (%) | Asian | 2 | 50 (1.9) | Ethnicity, n (%) | Asian | 0 | 1444 (1.9) |
| | Black/ African American | | 673 (25.4) | | Black/ African American | | 8100 (10.7) |
| | Native American or Alaska Native | | 3 (0.1) | | Native American or Alaska Native | | 561 (0.7) |
| | Native Hawaiian | | 1 (0.0) | | Hispanic | | 3116 (4.1) |
| | Other Pacific Islander | | 8 (0.3) | | | | |
| | Other/ Not Listed | | 603 (22.7) | | | | |
| | Patient Refused | | 16 (0.6) | | | | |
| | Unknown | | 25 (0.9) | | Other/ Unknown | | 3990 (5.3) |
| | White/ Caucasian | | 1267 (47.9) | | White/ Caucasian | | 58390 (77.2) |
| Insurance, n (%) | Medicaid | 0 | 492 (18.6) | | | | |
| | Medicare | | 891 (33.6) | | | | |
| | Private | | 1082 (40.9) | | | | |
| | Uninsured | | 183 (6.9) | | | | |

SOFA score between Black and White patients with COVID-19. In the eICU cohort, SOFA score was better than age at discriminating between survivors and non-survivors both in Black and White patients. Additionally, the SOFA score performed better in Black patients compared to White patients in the eICU cohort.

In the COVID-19 cohort, there was no significant difference in survival rates between Black and White patients at and above SOFA scores of 3 (S3 Fig). The survival rate of COVID-19 patients only dropped below 50% at SOFA scores greater than or equal to 12 (S4 Fig).

## Discussion

Age significantly outperformed SOFA score for predicting mortality in hospitalized COVID-19 patients, including those in the ICU. This phenomenon may be unique to COVID-19 as SOFA score was significantly better at predicting mortality in the eICU cohort. The finding

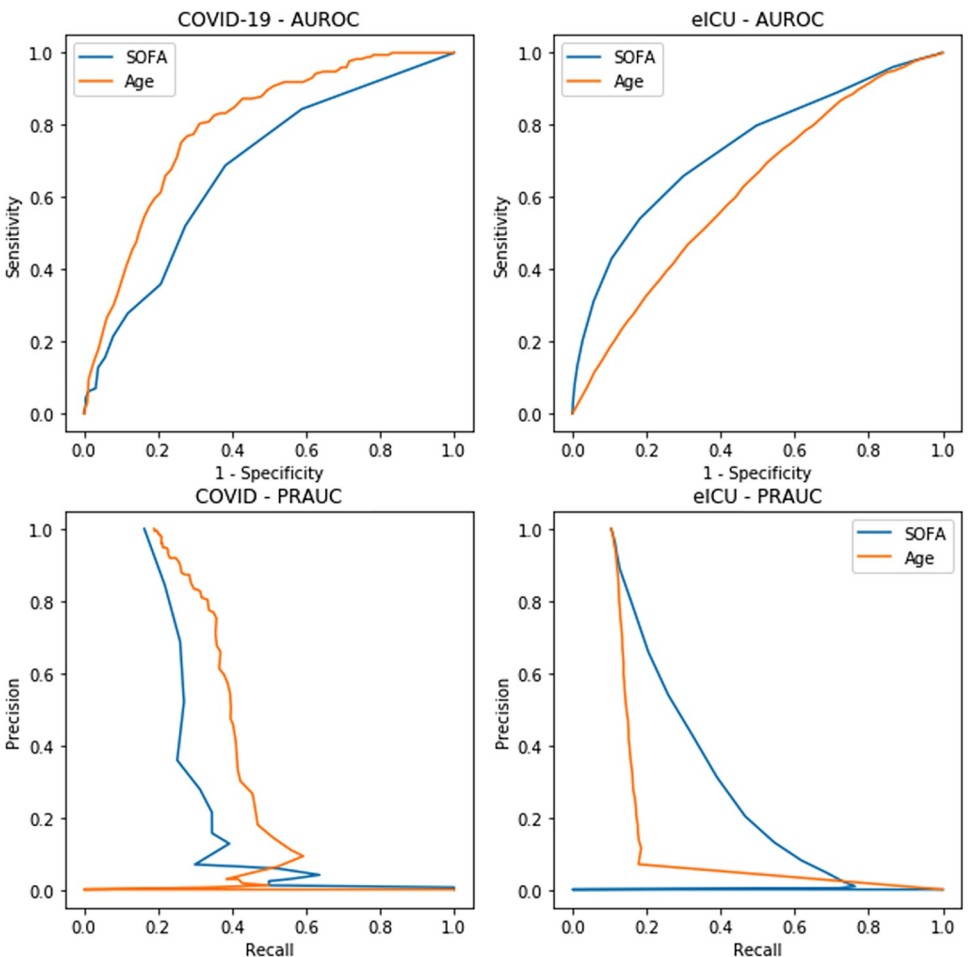

**Fig 2. AU-ROCs and AU-PRCs of age and SOFA score for mortality prediction.** Age-based model is in orange and SOFA score-based model is in blue.

that a simple metric such as age outperformed SOFA score for mortality prediction in COVID-19 patients suggests that caution should be taken when applying established prediction models to a completely novel disease process. This is especially prudent when using mortality prediction models to guide treatment decisions and resource allocation. Many guidelines for crisis standards of care suggested a SOFA score threshold of $\geq 6$ to identify patients with low likelihood of survival [29,30]. However, there was only a 33% mortality rate using that threshold in our COVID-19 cohort.

**Table 2. Model results.**

|  | Predictor | AU-ROC (95% CI) | AU-PRC |
|---|---|---|---|
| COVID-19 Cohort | Age | 0.795 (0.762, 0.828) | 0.387 |
|  | SOFA Score | 0.679 (0.638, 0.721) | 0.289 |
| COVID-19 ICU Sub-Cohort | Age | 0.698 (0.631, 0.765) | 0.455 |
|  | SOFA Score | 0.562 (0.486, 0.639) | 0.375 |
| eICU Cohort | Age | 0.628 (0.608, 0.628) | 0.177 |
|  | SOFA Score | 0.735 (0.726, 0.745) | 0.320 |

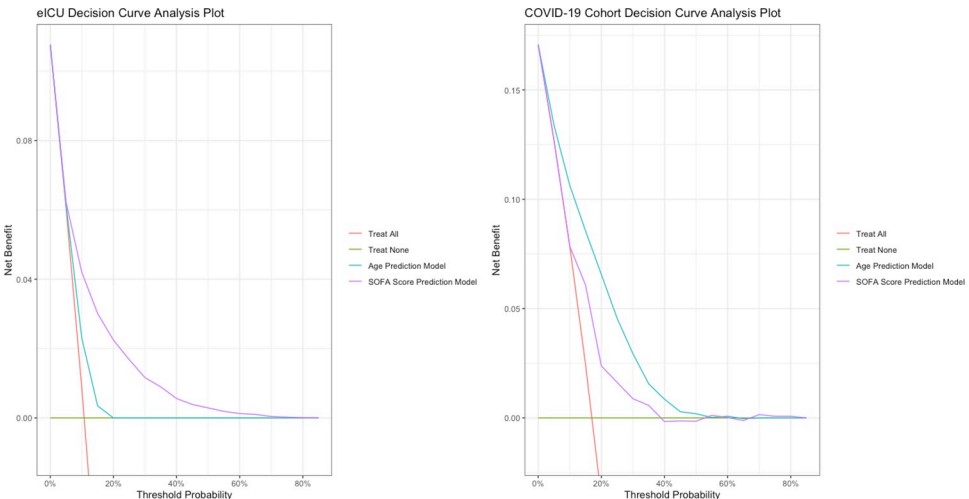

**Fig 3. Decision curve analysis plots of age-based and SOFA score-based models in the COVID-19 and eICU cohorts.** Decision curve analysis plots for age-based (blue) and SOFA score-based (purple) models in both cohorts. The "null" options of "treat all" and "treat none" are represented by red and green lines respectively.

Although the SOFA score was originally created for ICU patients [1], the inclusion of non-ICU patients in the COVID-19 cohort does not explain our findings. When restricted to COVID-19 cohort to ICU patients, a similar trend was noted. Age performed better than SOFA score for mortality prediction, with minimally overlapping confidence intervals. However, these results were limited by the relatively small sample size (642) of COVID-19 ICU

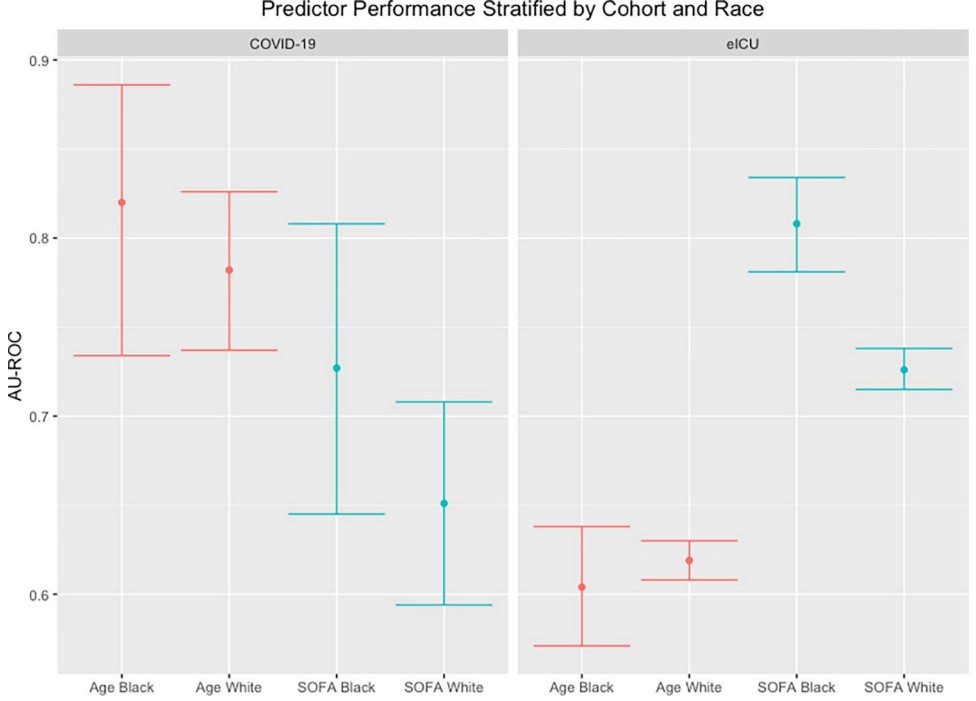

**Fig 4. Predictor performance stratified by cohort and race.** Plot of AU-ROCs with 95% CIs for age-based (red) and SOFA score-based (blue) models in each cohort stratified by race.

Table 3. AU-ROC of age and SOFA score for mortality prediction stratified by cohort and race.

| | AU-ROC (95% CI) | |
| --- | --- | --- |
| | **Black Patients** | **White Patients** |
| **COVID-19 Cohort** | | |
| Age | 0.82 (0.734, 0.886) | 0.782 (0.737, 0.826) |
| SOFA score | 0.727 (0.645, 0.808) | 0.651 (0.594, 0.708) |
| **eICU Cohort** | | |
| Age | 0.604 (0.571, 0.638) | 0.619 (0.608, 0.630) |
| SOFA score | 0.808 (0.781, 0.834) | 0.726 (0.715, 0.738) |

patients. While applying SOFA score to non-ICU patients is outside of the original intent of SOFA score, so is using it for mortality prediction. In the original study describing the SOFA score, Vincent et al. state, "it is important to realize that the SOFA score is designed not to *predict* outcome but to *describe* a sequence of complications"[1].

Prior studies have theorized that SOFA score underperforms for mortality prediction in COVID-19 patients because the illness affects fewer organ systems resulting in lower variability in scores [16]. Contradictory to that theory, patients in the COVID-19 cohort had the same standard deviation in scores (2.7) as the eICU cohort. Since age >65 is one of the characteristics with the strongest association with increased mortality in COVID-19 positive patients [31,32], it may partially explain why age outperforms SOFA score only in COVID-19 patients.

There was no difference between White and Black COVID-19 patients in the performance of either age or SOFA score for mortality prediction. Additionally, there was no significant difference in mortality rate between White and Black COVID-19 patients at SOFA scores greater than 2. Notably, a prior study of patients from the same COVID-19 cohort, Black patients had 1.5 times the odds of a SOFA score $\geq 6$ than white patients, even when adjusting for age, sex, insurance status, BMI, liver and renal diseases [23]. This suggests that Black patients with COVID-19 may be more likely than White patients to be assigned higher SOFA scores but will have similar mortality rates at those higher SOFA scores.

This study builds on a recent analysis from a single health system of 675 COVID-19 positive patients requiring mechanical ventilation in which SOFA score had inferior prognostic accuracy for in-hospital mortality compared to simply using patient age [16]. However, the study was limited by its cohort having higher and less variable SOFA scores than the average ICU population as well as missing data for approximately 25% of the cohort. Our study had over 2,600 COVID-19 positive patients with an average SOFA score similar to prior analyses on non-COVID-19 patients [6,7] and a mortality rate comparable to other cohorts of COVID-19 patients that were admitted to U.S. hospitals during a similar time period [31]. Moreover, the SOFA scores used in the COVID-19 cohort were automatically calculated by the electronic health record, a pragmatic approach that would be used in triage scenarios.

Our study has several limitations. The cohort of COVID-19 patients was restricted to a single health system in the northeast U.S. which may not have a comparable population to COVID-19 patients at other academic health systems or those in the eICU database. Due to sample size, we also restricted our analysis to two racial groups and did not consider patients that identified as multi-racial. We did not consider sex or its potential interaction with race or age in our study. Furthermore, an unknown proportion of patients in the COVID-19 cohort were too critically ill to answer demographic questions and had their race and sex recorded by a hospital clerk based on assumption. Additionally, this study only describes the performance of max SOFA score within 24 hours of hospital admission. Many COVID-19 positive patients present to the hospital with respiratory complaints and develop multisystem organ

dysfunction later in their disease course [32]. However, information on SOFA subscores in the cohort of COVID-19 patients were not available so we were unable to test this hypothesis. SOFA score may have greater utility for predicting mortality with serial measurements or later in disease course [2].

Although COVID-19's status as an official U.S. public health emergency ended May 11, 2023 [33], hospital system capacity continues to be strained by both the direct and in-direct effects of COVID-19 [34,35]. Nationwide, there remains key gaps in planning for crisis standards of care, and SOFA score is often proposed as a guide for resource allocation [36]. This study suggests caution should be used when considering SOFA score as triage tool, as it has limited prognostic performance in COVID-19 and may suffer similar limitations in future pandemics, especially when novel disease processes are involved.

## Supporting information

**S1 Fig. Calibration curves for SOFA score and age based models in the COVID-19 and eICU cohorts.** Calibration curves for age-based (red) and SOFA score-based (blue) models in both cohorts. The black line with a slope of 1 represents a perfectly calibrated model.
(TIF)

**S2 Fig. AU-ROCs of age and SOFA score for mortality prediction in COVID-19 cohort patients admitted to the ICU.** AU-ROC of age-based (orange) and SOFA score-based (blue) models calculated on subset of COVID-19 cohort that required ICU admission.
(TIF)

**S3 Fig. Mortality in COVID-19 patients above sofa score threshold by race with 95% CIs.** Mortality percentage at and above a given SOFA score in Black (red line) and White (blue line) patients in the COVID-19 cohort with 95% confidence intervals in shaded region.
(TIF)

**S4 Fig. Mortality in COVID-19 patients above SOFA score threshold.** Mortality percentage at and above a given SOFA score for all patients in the COVID-19 cohort.
(TIF)

## Author Contributions

**Conceptualization:** Raphael A. G. Sherak, Naveed Khimani, Benjamin Tolchin, Karen Jubanyik, Bobak J. Mortazavi, Adrian D. Haimovich.

**Data curation:** Raphael A. G. Sherak, Hoomaan Sajjadi, Naveed Khimani, Adrian D. Haimovich.

**Formal analysis:** Raphael A. G. Sherak, Hoomaan Sajjadi, Naveed Khimani, Bobak J. Mortazavi, Adrian D. Haimovich.

**Investigation:** Raphael A. G. Sherak, Hoomaan Sajjadi, Naveed Khimani, Adrian D. Haimovich.

**Methodology:** Raphael A. G. Sherak, Karen Jubanyik, R. Andrew Taylor, Wade Schulz, Bobak J. Mortazavi, Adrian D. Haimovich.

**Project administration:** R. Andrew Taylor, Adrian D. Haimovich.

**Resources:** Benjamin Tolchin, Karen Jubanyik, R. Andrew Taylor, Wade Schulz, Bobak J. Mortazavi, Adrian D. Haimovich.

**Supervision:** Benjamin Tolchin, Karen Jubanyik, R. Andrew Taylor, Wade Schulz, Bobak J. Mortazavi, Adrian D. Haimovich.

**Validation:** Raphael A. G. Sherak, Adrian D. Haimovich.

**Visualization:** Raphael A. G. Sherak, Adrian D. Haimovich.

**Writing – original draft:** Raphael A. G. Sherak.

**Writing – review & editing:** Raphael A. G. Sherak, Hoomaan Sajjadi, Naveed Khimani, Benjamin Tolchin, Karen Jubanyik, R. Andrew Taylor, Wade Schulz, Bobak J. Mortazavi, Adrian D. Haimovich.

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
