## [Decision Letter · Decision Letter 0]

23 Nov 2023

PONE-D-22-20001SOFA score performs worse than age for predicting mortality in patients with COVID-19PLOS ONE

Dear Dr. Sherak,

Thank you for submitting your manuscript to PLOS ONE. After careful consideration, we feel that it has merit but does not fully meet PLOS ONE’s publication criteria as it currently stands. Therefore, we invite you to submit a revised version of the manuscript that addresses the points raised during the review process.

We look forward to receiving your revised manuscript.

Kind regards,

Dong Wook Jekarl

Academic Editor

PLOS ONE

Journal Requirements:

 "The author(s) received no specific funding for this work.

BT receives research support from the US Department of Veterans Affairs (https://www.newengland.va.gov/research/v1cda/) and the C.G. Swebilius Foundation (https://fconline.foundationcenter.org/fdo-grantmaker-profile/?key=SWEB001). "

Reviewers' comments:

Reviewer's Responses to Questions

**Comments to the Author**

1. Is the manuscript technically sound, and do the data support the conclusions?

Reviewer #1: Yes

Reviewer #2: Yes

2. Has the statistical analysis been performed appropriately and rigorously? 

Reviewer #1: Yes

Reviewer #2: No

3. Have the authors made all data underlying the findings in their manuscript fully available?

Reviewer #1: Yes

Reviewer #2: Yes

4. Is the manuscript presented in an intelligible fashion and written in standard English?

Reviewer #1: Yes

Reviewer #2: Yes

5. Review Comments to the Author

Reviewer #1: The authors performed a retrospective study to evaluate whether SOFA score is desirable in predicting in-hospital mortality of COVID-19 patients and compared it with simple use of age. The authors concluded age is better than SOFA score. Although using age for predicting mortality of COVID-19 patients is still far from perfect, the information derived from this study is very important to the field. The methods used in the study was straightforward, I have no criticism on the analysis.

Reviewer #2: It is a very interesting study. "Caution is needed when applying an established disease severity index model to a new illness." A few minor issues warrant attention:

1.In the "methods" section of the article, it is stated: "Age in years at time of admission and the maximum SOFA score recorded in the first 24 hours after admission were each used as predictor variables in separate univariate logistic regression models for the binary outcome of in-hospital mortality. Each model was fit to a sample of 60% of the respective cohorts." However, in the “results”section, relevant displays of these outcomes were not observed.

2.It is imperative to present the results where age, after adjusting for common confounding factors (comorbidities, creatinine, total bilirubin, INR, etc.) in both cohorts, is independently associated with prognosis. Additionally, for the SOFA score, the prognostic predictive value after adjusting for confounding factors should also be demonstrated.

3.Obviously, in this study, the author aims to assess the discriminative performance of SOFA scores on patient mortality and compare it with age. So, in addition to comparing the differences in AUC (statistical test results were not provided in the article), a comprehensive judgment needs to be made by combining NRI and IDI.

6. PLOS authors have the option to publish the peer review history of their article (what does this mean?). If published, this will include your full peer review and any attached files.

Reviewer #1: No

Reviewer #2: No

---

## [Author Response · Author response to Decision Letter 0]

7 Jan 2024

Below is a copy of our uploaded response to reviewer comments

Response to PLOS One Reviewer Comments

1/5/2023

We would like to thank the Academic Editor and Reviewers for their thoughtful comments and suggestions. 

Our point-by-point response is formatted in italics and red below each comment for context 

Response to Editor’s Comments

Thank you. We have updated the formatting of our manuscript to meet PLOS ONE’s style requirements.

Unfortunately, we are unable to publicly share the data from the COVID cohort (see data availability statement below. However, the eICU cohort is taken from the publicly available eICU dataset: Pollard, T., Johnson, A., Raffa, J., Celi, L. A., Badawi, O., & Mark, R. (2019). eICU Collaborative Research Database (version 2.0). PhysioNet. https://doi.org/10.13026/C2WM1R.

 "The author(s) received no specific funding for this work.

BT receives research support from the US Department of Veterans Affairs (https://www.newengland.va.gov/research/v1cda/) and the C.G. Swebilius Foundation (https://fconline.foundationcenter.org/fdo-grantmaker-profile/?key=SWEB001). 

 We did not receive any funding (financial or material) for this study.

 There were no funders for this study. 

 While the authors received no specific funding for this work or study, BT receives salary and research support for other projects from the US Department of Veterans Affairs (https://www.newengland.va.gov/research/v1cda/) and the C.G. Swebilius Foundation (https://fconline.foundationcenter.org/fdo-grantmaker-profile/?key=SWEB001). These funders had no involvement in the study. 

The authors received no specific funding for this work.

 Thank you. We have added the following to the box underneath the “Data Availability” section in the submission form:

“We have discussed the sharing of our data with the Yale University Privacy Office which made the determination that we are legally and ethically restricted from sharing data because the extent of data poses a risk of re-identification of patients and their HIPAA protected data through deductive disclosure.

 Susan Bouregy, PhD (susan.buregy@yale.edu) is Yale’s chief privacy officer, and will serve as the contact for the Yale University Privacy Office, to which data requests may be sent.

Data cannot be shared publicly because of legal and ethical restrictions regarding patient privacy. Data are available from Yale's chief privacy officer, Susan Bouregy, PhD (susan.buregy@yale.edu), for researchers who meet the criteria for access to confidential data.

This was the same case for “Tolchin B, Oladele C, Galusha D, Kashyap N, Showstark M, Bonito J, et al. (2021) Racial disparities in the SOFA score among patients hospitalized with COVID-19. PLoS ONE 16(9): e0257608. https://doi.org/10.1371/journal.pone.0257608” which used the same dataset and was previously published in your journal. 

The eICU database is publicly accessible. Pollard, T., Johnson, A., Raffa, J., Celi, L. A., Badawi, O., & Mark, R. (2019). eICU Collaborative Research Database (version 2.0). PhysioNet. https://doi.org/10.13026/C2WM1R.

1. Is the manuscript technically sound, and do the data support the conclusions?

Reviewer #1: Yes

Reviewer #2: Yes

2. Has the statistical analysis been performed appropriately and rigorously?

Reviewer #1: Yes

Reviewer #2: No

3. Have the authors made all data underlying the findings in their manuscript fully available?

Reviewer #1: Yes

Reviewer #2: Yes

4. Is the manuscript presented in an intelligible fashion and written in standard English?

Reviewer #1: Yes

Reviewer #2: Yes

5. Review Comments to the Author

Reviewer #1: The authors performed a retrospective study to evaluate whether SOFA score is desirable in predicting in-hospital mortality of COVID-19 patients and compared it with simple use of age. The authors concluded age is better than SOFA score. Although using age for predicting mortality of COVID-19 patients is still far from perfect, the information derived from this study is very important to the field. The methods used in the study was straightforward, I have no criticism on the analysis.

We thank reviewer one for this positive feedback about our manuscript and taking the time to review it. 

Reviewer #2: It is a very interesting study. "Caution is needed when applying an established disease severity index model to a new illness." A few minor issues warrant attention:

1.In the "methods" section of the article, it is stated: "Age in years at time of admission and the maximum SOFA score recorded in the first 24 hours after admission were each used as predictor variables in separate univariate logistic regression models for the binary outcome of in-hospital mortality. Each model was fit to a sample of 60% of the respective cohorts." However, in the “results”section, relevant displays of these outcomes were not observed.

We additionally thank this reviewer for their interest in our manuscript.

We would like to take this opportunity to provide clarification for the “relevant displays” requested by the reviewer. In this work, we develop univariate logistic regression models of COVID-19 mortality risk. In order to assess the accuracy of the univariate logistic regression model, we use area under the receiver operating characteristic curve (AU-ROC), area under the precision-recall curves (AU-PRC). These are found in Figure 2. Based on this reviewer’s feedback, we now include calibration curves which are important to understanding the clinical utility of a prognostic model. These additional analyses are reflected in the methods section and the calibration curve figures are included as the new Supplemental Figure 1 with observed probability of event (mortality) on the Y axis and predicted event rate (mortality) on the X axis.[1] 

2.It is imperative to present the results where age, after adjusting for common confounding factors (comorbidities, creatinine, total bilirubin, INR, etc.) in both cohorts, is independently associated with prognosis. Additionally, for the SOFA score, the prognostic predictive value after adjusting for confounding factors should also be demonstrated.

We thank this reviewer for this feedback. Our goal was to describe the utility of these measures as strictly univariate screening tools, rather than in a more complex algorithm. Moreover, two proposed confounders (creatinine, bilirubin) are components of SOFA. We are unsure of how we can adequately control for these potential confounders when they are part of the exposure.

3.Obviously, in this study, the author aims to assess the discriminative performance of SOFA scores on patient mortality and compare it with age. So, in addition to comparing the differences in AUC (statistical test results were not provided in the article), a comprehensive judgment needs to be made by combining NRI and IDI.

To these author’s understanding, NRI and IDI refer to the change in performance once a second test is added to a baseline measure.[2] In our study, there is no baseline measure, but instead a head-to-head comparison of two univariate measures (age and SOFA score). We instead added a decision curve analysis (DCA) as a new Figure 3 to specifically address tension related to clinical decision making.[3] Consistent with earlier findings in our paper, the age based prediction model has a higher net benefit than the SOFA Score based model across a large range of threshold probabilities in the COVID cohort, while the opposite phenomena is observed in the eICU (non-COVID) cohort. 

1. Kuhn M, Vaughan D, Ruiz E. probably: Tools for Post-Processing Class Probability Estimates. 2023. Available: https://github.com/tidymodels/probably/, https://probably.tidymodels.org}

2. Hilden J. Commentary: On NRI, IDI, and “good-looking” statistics with nothing underneath. Epidemiology . 2014. pp. 265–267. doi:10.1097/EDE.0000000000000063

3. Fitzgerald M, Saville BR, Lewis RJ. Decision curve analysis. JAMA: the journal of the American Medical Association. 2015. pp. 409–410. doi:10.1001/jama.2015.37

---

## [Decision Letter · Decision Letter 1]

11 Mar 2024

SOFA score performs worse than age for predicting mortality in patients with COVID-19

PONE-D-22-20001R1

Dear Dr. Sherak,

We’re pleased to inform you that your manuscript has been judged scientifically suitable for publication and will be formally accepted for publication once it meets all outstanding technical requirements.

Kind regards,

Chiara Lazzeri

Academic Editor

PLOS ONE

Additional Editor Comments (optional):

Reviewers' comments:

Reviewer's Responses to Questions

**Comments to the Author**

1. If the authors have adequately addressed your comments raised in a previous round of review and you feel that this manuscript is now acceptable for publication, you may indicate that here to bypass the “Comments to the Author” section, enter your conflict of interest statement in the “Confidential to Editor” section, and submit your "Accept" recommendation.

Reviewer #1: All comments have been addressed

Reviewer #2: All comments have been addressed

2. Is the manuscript technically sound, and do the data support the conclusions?

Reviewer #1: Yes

Reviewer #2: Yes

3. Has the statistical analysis been performed appropriately and rigorously? 

Reviewer #1: Yes

Reviewer #2: Yes

4. Have the authors made all data underlying the findings in their manuscript fully available?

Reviewer #1: Yes

Reviewer #2: Yes

5. Is the manuscript presented in an intelligible fashion and written in standard English?

Reviewer #1: Yes

Reviewer #2: Yes

6. Review Comments to the Author

Reviewer #1: The authors have adequately addressed your comments raised in a previous round of review. I have no further questions.

Reviewer #2: All comments have been addressed, and no further question.

7. PLOS authors have the option to publish the peer review history of their article (what does this mean?). If published, this will include your full peer review and any attached files.

Reviewer #1: No

Reviewer #2: No

---

## [Editor Report · Acceptance letter]

8 May 2024

PONE-D-22-20001R1 

PLOS ONE

Dear Dr. Sherak, 

I'm pleased to inform you that your manuscript has been deemed suitable for publication in PLOS ONE. Congratulations! Your manuscript is now being handed over to our production team.

Kind regards, 

on behalf of

Dr. Chiara Lazzeri 

Academic Editor

PLOS ONE